# Structures of pseudorabies virus capsids

Guosong Wang[1,2,4], Zhenghui Zha[1,2,4], Pengfei Huang[1,2,4], Hui Sun[1,2,4], Yang Huang[1,2], Maozhou He [1,2], Tian Chen[1,2], Lina Lin[1,2], Zhenqin Chen[1,2], Zhibo Kong[1,2], Yuqiong Que[1,2], Tingting Li [1,2], Ying Gu [1,2], Hai Yu[1,2], Jun Zhang [1,2], Qingbing Zheng [1,2✉], Yixin Chen [1,2✉], Shaowei Li [1,2✉] & Ningshao Xia [1,2,3✉]

Pseudorabies virus (PRV) is a major etiological agent of swine infectious diseases and is responsible for significant economic losses in the swine industry. Recent data points to human viral encephalitis caused by PRV infection, suggesting that PRV may be able to overcome the species barrier to infect humans. To date, there is no available therapeutic for PRV infection. Here, we report the near-atomic structures of the PRV A-capsid and C-capsid, and illustrate the interaction that occurs between these subunits. We show that the C-capsid portal complex is decorated with capsid-associated tegument complexes. The PRV capsid structure is highly reminiscent of other α-herpesviruses, with some additional structural features of β- and γ-herpesviruses. These results illustrate the structure of the PRV capsid and elucidate the underlying assembly mechanism at the molecular level. This knowledge may be useful for the development of oncolytic agents or specific therapeutics against this arm of the herpesvirus family.

[1] State Key Laboratory of Molecular Vaccinology and Molecular Diagnostics, School of Public Health, School of Life Sciences, Xiamen University, Xiamen 361102, China. [2] National Institute of Diagnostics and Vaccine Development in Infectious Diseases, Xiamen University, Xiamen 361102, China. [3] Research Unit of Frontier Technology of Structural Vaccinology, Chinese Academy of Medical Sciences, Xiamen 361102, China. [4] These authors contributed equally: Guosong Wang, Zhenghui Zha, Pengfei Huang, Hui Sun. ✉email: abing0811@xmu.edu.cn; yxchen2008@xmu.edu.cn; shaowei@xmu.edu.cn; nsxia@xmu.edu.cn

The pseudorabies virus (PRV) is the causative agent of Aujeszky's disease, which leads to severe economic losses in the swine industry worldwide[1–3]. Pigs are the natural host and reservoir of PRV, with PRV infection characterized by fatal encephalitis, respiratory distress, a block in the growth of growing and fattening pigs, reproductive failure in sows, and 100% mortality in newborn piglets[4]. PRV can also infect numerous other mammals except for primates, such as ruminants, rodents, and carnivores, with nearly 100% mortality[5,6]. Despite this, accumulative evidence suggests that PRV may be a potential threat to humans, given that glycoprotein D (gD) of PRV can bind to human nectin-1[7]. Furthermore, there have been several cases of human viral encephalitis caused by PRV infection[8–13]. Indeed, a recent study from nine provinces in China detected PRV antibody-positive rates in patients with encephalitis of 12.16%, 14.25%, and 6.52% in 2012, 2013, and 2017, respectively; this compares with just 3.86% in healthy people[14]. This potential hazard of PRV in humans has so far remained somewhat neglected despite warranting further investigation.

PRV is a member of the alpha-herpesvirinae subfamily and the *Varicellovirus* genus[15]. PRV has a double-stranded genomic DNA of approximately 143 kb and contains at least 72 genes[16]. PRV is an ideal model not only for mechanistic investigations into α-herpesvirus neurotropism but also for the structural determination of herpesvirus capsids. The high-resolution structures of viral capsids from many human herpesviruses, e.g., Herpes Simplex Virus Type 1 (HSV-1), HSV-2, varicella-zoster virus (VZV), human cytomegalovirus (HCMV), Human Herpesvirus 6 (HHV-6), Kaposi's sarcoma associated herpesvirus (KSHV), and Epstein-Barr virus (EBV), have been resolved, as has the precise portal structure of HSV-1, KSHV, and EBV[17–27]. However, only medium-resolution structures of PRV capsids have been shown to date[28–30]. The PRV capsid, similar to other herpesviruses, has a triangulation (T) number of 16 and is composed of major capsid proteins (MCPs), small capsid proteins (SCPs), and triplexes. The MCP (VP5) is encoded by UL19 and forms pentamer (penton) and hexamer (hexon) in the capsid. The hexon MCP is decorated with the SCP (VP26 protein, encoded by UL35), whereas the penton MCP lacks decoration. The viral capsid thus contains a total of 150 hexon MCPs, 150 SCPs and a total of 11 penton MCPs. As shown for other herpesviruses[26,27,31], the twelfth vertex of the icosahedral PRV capsid is presumed to encompass a special cylindrical channel called a portal, which is formed by 12 portal proteins encoded by UL6. Through this portal is where the PRV packages and releases the genome. The triplexes, each composed of one VP19C protein (Tri1, encoded by UL38) and two VP23 proteins (Tri2, encoded by UL18), anchor to the capsid floor via Tri1 N-anchor. Triplexes can be divided into six types (Ta, Tb, Tc, Td, Te, Tf) in terms of their localization on viral capsid. Thus, each mature PRV capsid contains a total of 955 pUL19 molecules, 900 pUL35 molecules, 640 pUL18 molecules, 320 pUL38 molecules, and 12 pUL6 molecules.

Here, we use cryo-EM to image both the PRV empty capsid resulted from abortive DNA packing (A-capsid) and mature capsids fully packed with DNA (C-capsid). Through localized reconstruction, we obtain the near-atomic structures of both capsids as well as the portal architecture of the C-capsid. These high-resolution structures illustrate the interaction of the PRV capsid subunits at the molecular level, and reveal the portal complex structure of the PRV C-capsid. This knowledge substantially expands the current structural understanding of herpesvirus family members, provides important information for the design of vaccines against all herpesvirus family members, and offers insight into potential therapeutics.

## Results

**Overall structures of PRV C-capsid and A-capsid.** We first sought to visualize virus particle assembly in cells. Panc-1 cells were infected with PRV for 24 h and then imaged by transmission electron microscopy (TEM) to observe the morphology and localization of PRV particles in thin sections. Typical A-capsids, C-capsids, and PRV virions could be observed in the infected cells (Supplementary Fig. 1a). The virus titer in the supernatant could reach to about $5 \times 10^7$ pfu/mL, after cell infected for 3 days. Thus, the supernatant was collected to purify capsid particles. The analytical ultracentrifugation was performed to resolve the particle composition of purified PRV particles (Supplementary Fig. 2). Three types of viral capsids with sedimentation coefficient of 750 S, 939 S, and 1340 S were resolved, which could be ascribe to PRV A-, B- and C-capsid, respectively, according to the previously studies[18]. Of these particles, the C-capsid is the dominant component whereas only a small fraction of A-capsids and few B-capsids were observed (Supplementary Fig. 2). The purified PRV capsids were then imaged by cryo-EM on a Thermo Fisher TF30 TEM equipped with a Falcon 3 detector, with data collection carried out in linear mode without electron counting (Supplementary Fig. 1b). From the same dataset, we selected a total of 14,252 and 8899 particles for PRV C-capsid and A-capsid (Supplementary Fig. 1c-d), respectively, and, by imposing icosahedral symmetry, obtained three-dimensional (3D) reconstructions at resolutions of 4.43 and 4.53 Å, respectively (Supplementary Fig. 3a-b, Supplementary Table 1). The 4 to 5-Å 3D reconstructions show a highly similar T = 16 icosahedral architecture for the C- and A-capsids, constituted by a capsomeric composition of 12 pentons, 150 hexons, 320 triangular triplexes, and the additional capsid-associated tegument complexes (CATCs), which were only present in the C-capsid not in the A-capsid (Fig. 1a-b).

We next sought to further improve the reconstruction resolutions and overcome the limitations of weakened densities induced by the enormous size of the viral capsid. To this end, we used a sub-particle reconstruction strategy to obtain local density maps of both the C- and A-capsids at near atomic resolutions. The 5-, 3-, and 2-fold sub-particles of both the C- and A-capsids were individually extracted, correcting for the local defocus parameters during extraction. Local reconstructions were then carried out to finally obtain the cryo-EM structures of the C-capsid 5-, 3-, and 2-fold sub-regions at 3.31, 3.46, and 3.43 Å, respectively (Fig. 1c-e, Supplementary Fig. 3a, c-e and Supplementary Table 1), and the A-capsid 5-, 3-, and 2-fold sub-regions at 3.64, 3.50, and 3.41 Å, respectively (Fig. 1f-h, Supplementary Fig. 3b, f-h and Supplementary Table 1).

The near-atomic resolution reconstructions allowed us to build the atomic models of both the C- and A-capsids (Fig. 2a). The backbone of the polypeptide and many of the side chains are clearly defined (Fig. 2c and Supplementary Figs. 4–6). Similar to most reported herpesvirus capsid structures, the structural components within one asymmetric unit of the PRV capsid contained the following organization: 16 copies of the major capsid proteins (MCP, VP5); one-fifth of a penton capsomer; 2 and a half hexon capsomers (1 P-hexon, 1 C-hexon, and half of E-hexon), each hexon MCP decorated with one copy (total 15 copies) of the small capsid protein (SCP, VP26); and 5 triplexes (Ta, Tb, Tc, Td, Te) along with one-third of a Tf triplex, each consisting of two conformers of the Tri2 VP23 and one of the Tri1 VP19C (Fig. 2a, b). Because the Tf triplex is located at the icosahedral 3-fold axis, the densities were smeared during 3D reconstruction with imposed icosahedral symmetry. The models of the asymmetric units derived from icosahedral reconstructions of the A- and C-capsids are nearly identical, with a root mean square deviation of 0.298 Å over 26,091 atoms, with the Cα positions of the two models aligned (Supplementary Fig. 7).

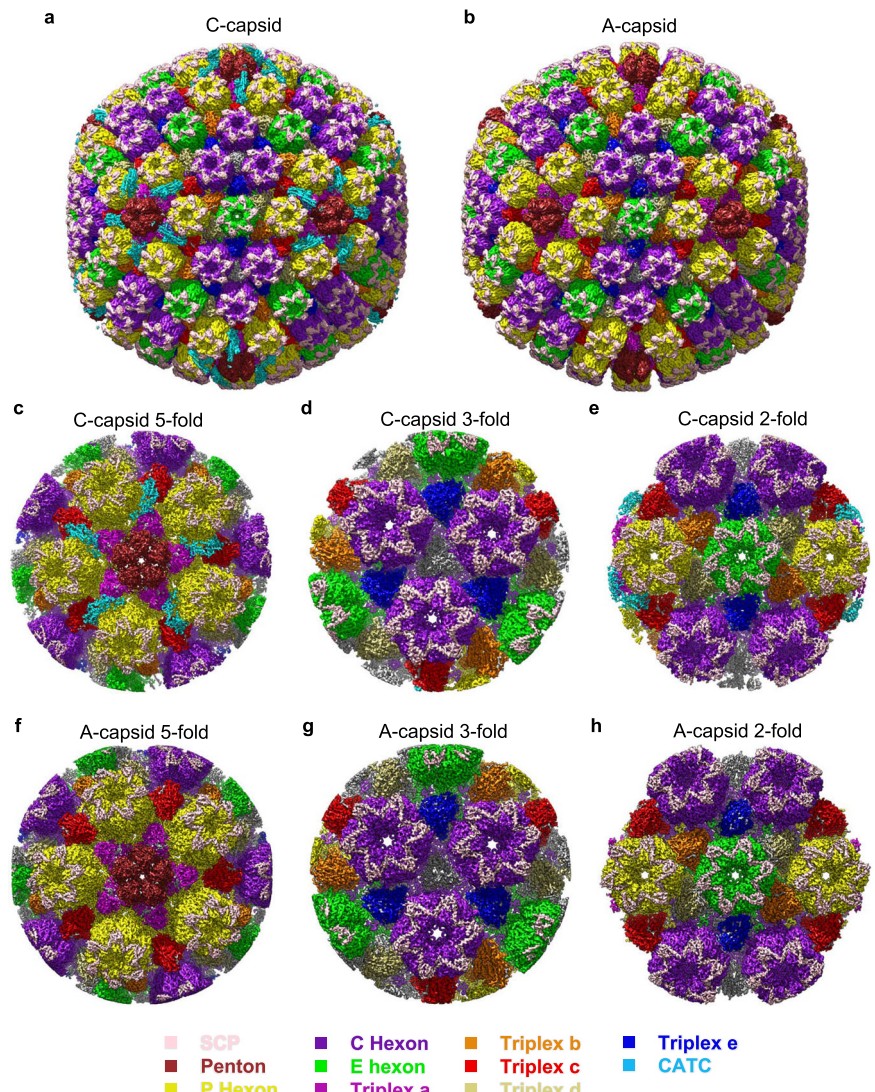

**Fig. 1 Icosahedral and sub-particle reconstructions of PRV C- and A-capsids. a**, **b** Shaded-surface representations of the PRV C-capsid (**a**) and A-capsid (**b**) revealing the presence of capsid-associated tegument complexes (CATCs) on the C-capsid. **c–e** Reconstructions for sub-particles extracted for the 5-fold (**c**), 3-fold (**d**), and 2-fold (**e**) of the C-capsid, respectively. **f–h** Reconstructions for sub-particles extracted for the 5-fold (**f**), 3-fold (**g**), and 2-fold (**h**) of the A-capsid, respectively.

**Structure of the MCP and interactions between MCPs**. The penton and hexon are composed mainly of MCPs, which are 1330 amino acids (aa) in length and about 150 kDa in molecular weight. MCPs are encoded by the UL19 gene, and are among the most conserved of all structural protein genes among herpesviruses (Supplementary Table 2). Indeed, similar to most other reported herpesviruses[21], the MCP of PRV is structurally conserved and can be divided into seven domains: N-lasso (aa 1-58), Johnson-fold (aa 59-182, 220-277, 351-383, 1,020-1,097), helix-hairpin (aa 184-215), dimerization (aa 276-350), buttress (aa 1,100-1,296), channel (aa 383-454, 1,297-1,323) and an upper domain (aa 455-1,016) (Fig. 3a). The Johnson-fold domain—so named after the characteristic folds first observed in phage HK97 gp5[32]—is present in many DNA phages and in MCPs of herpesviruses. Superimposition of all 15 copies of hexon MCPs from one asymmetric unit (C1-C6, P1-P6, and E1-E3) show their highly structure similarity but the P1 and P6 have slightly different conformations in the N-lasso and dimerization domains (Supplementary Fig. 8a-c), which is consistent to previously observation in HSV-2 and EBV[18,25,33]. In addition, the penton and hexon MCPs also shows high similarity at the tower section

but minor conformational diversity on the buttress domain (Supplementary Fig. 8d,e) and major structural deviations on the capsid floor, particularly at the N-lasso dimerization and Johnson-fold (Fig. 3b). Hexons and pentons also show slight discrepancies in their central channel size, with diameters of 13 Å and 11 Å, respectively (Fig. 3c); both channel sizes are insufficient for the translocation of double-stranded DNA (~22 Å in diameter).

Similar to other herpesviruses, there are potentially three types of MCP-MCP interactions that can occur between hexon MCPs, which are mediated by the floor regions (Fig. 3d-g). The type I interaction is intra-capsomeric, formed between two adjacent MCPs via the β-strands or loops of the Johnson-fold and E-Loop domains of P3, the β-strands of the N-lasso of C5, and P2; a stable two-stranded β-sheets and two adjacent loops form interaction network by hydrogen bonding among these segments (Fig. 3g). The type II interaction, comparatively, involves the dimerization of P3 and C6 MCPs located adjacent to the 2-fold region (Fig. 3f). Finally, in the type III interaction, the N-lasso domain of C5 extends to the position of the P1 and P2 MCPs, and surrounds the β-sheet in the type I interaction, forming a so-called "lasso"

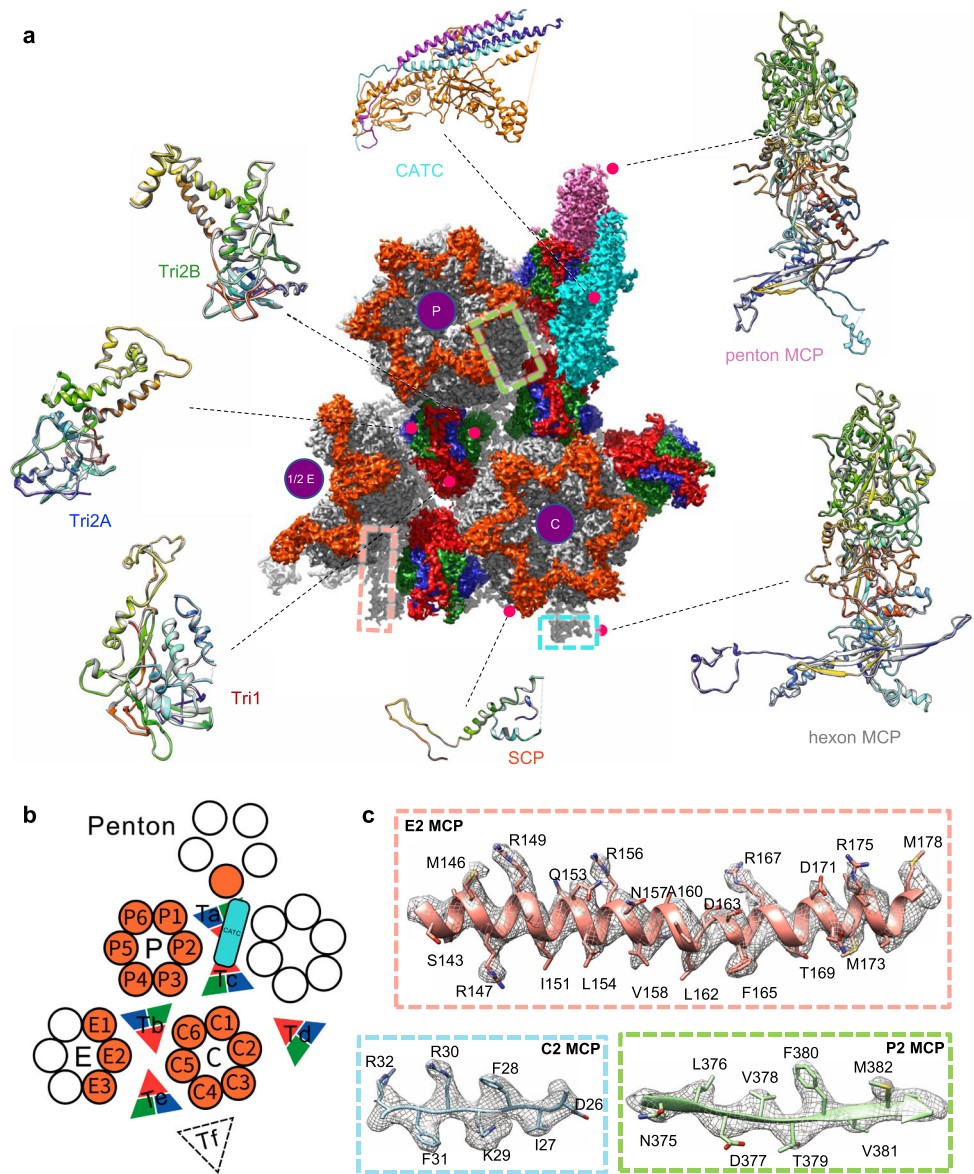

**Fig. 2 Atomic models of the PRV C- and A-capsids. a** Density map of an icosahedral asymmetric unit of the C-capsid, segmented from the three main-axis sub-particle reconstructions and colored by protein type: MCP (gray), Tri1 (red), Tri2A (blue), Tri2B (green), SCP (orange-red), and CATC (cyan). Representative superimposed atomic models of protein subunits from C-capsid (rainbow-colored from blue to red) and A-capsid (gray) are shown next to the density map as ribbons. **b** A schematic representation of above asymmetric unit (shaded) of the capsid. **c** Segmented density maps (mesh) and corresponding atomic models of MCP which illustrate side chain features. Residues with side chains are labeled. Abbreviations: CATC Capsid-associated tegument complexes; MCP Major capsid protein; SCP Small capsid protein; Tri Triplex.

interaction (Fig. 3f-g); the N-terminus of P2 tethers the β-sheets of C5 and C6. This type III interaction builds on the type I interaction and consolidates the type II interaction, fortifying capsid stability.

However, because of loose and disordered N-lasso and dimerization domains of the penton MCP, interactions between the penton MCP and the adjacent hexon (P1 and P6) MCPs were relatively weak and none of the three classical MCP-MCP interactions were observed (Fig. 3e and Supplementary Fig. 9). Such weak interactions between the penton and hexon MCPs were reported previously for varicella-zoster virus (VZV)[19]. Comparatively, in the other two representative α-herpesvirus members, HSV-1 and HSV-2, the dimerization region of the α-domain of the penton and P6 MCP is refolded to form a strong interaction mediated by an antiparallel double-helix. In order to compare the capsid stabilities of PRV and HSV-1, we performed

the thermal stability assay and the result showed that HSV-1 capsid withstood high temperature of up to 74 °C, which is similar to the previous studies[34,35], and significantly higher than that of PRV capsids (~53 °C) (Supplementary Fig. 10). This result indicates that the capsid of PRV is less stable than that of HSV-1 possibly due to its weaker inter-subunit interactions.

**Structure of triplex and its interaction with MCP.** There are 320 triplexes on the PRV capsid, which inlay in 3-fold or quasi 3-fold regions where the MCP N-lasso triangles interact with each other (Fig. 4a). A triplex is a heterotrimer consisting of one unique Tri1 conformer and two Tri2 conformers (Tri2A and Tri2B) in the form of a Tri2 dimer (Fig. 4b-d). The "embracing" arms between Tri2A and Tri2B in the dimer adopt different conformations in the two conformers (Fig. 4c). Tri1 in PRV is 368 aa in length, and

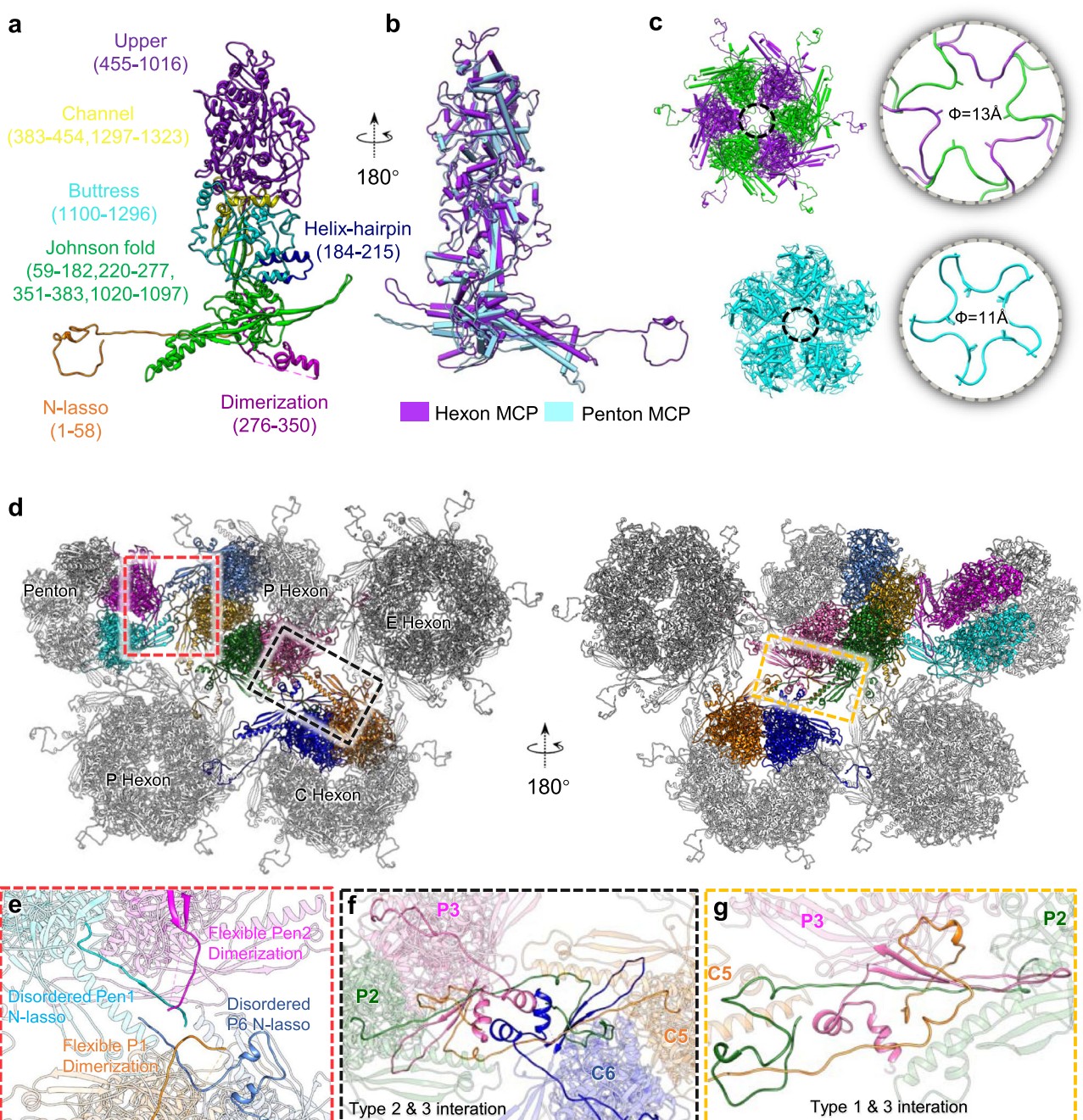

**Fig. 3 MCP structure and MCP-MCP interactions. a** Canonical hexon MCP (C5 hexon MCP) that colored and labeled by domain. **b** Superposition of the hexon and penton MCPs. **c** Pipe-and-plank depiction of an E-hexon and a penton showing the central channel (dashed circular). The narrowest regions of the hexon (top) and penton (below) have a dimension of about 11–13 Å. **d** Part of the MCP network viewed from outside (left) and inside (right). **e** Interactions between hexon and penton MCPs. **f**, **g** Interactions between hexon MCPs (type I, II, and III interactions; see main text). Abbreviation, MCP Major capsid protein.

about 100 aa shorter than that in other α-herpesviruses (HSV-1, 465 aa, HSV-2, 466 aa, and VZV, 483 aa; Supplementary Fig. 11a). The most obvious difference lies in the N-terminal anchoring sequence of Tri1, which is extremely short (23 aa); the gene sequence resembles that of β-herpesviruses rather than other α-herpesviruses, which have long N-terminal anchored domains (more than 100 aa) (Supplementary Fig. 11a). The short N-terminus of PRV Tri1 serves as an anchored domain extending into the inner side of capsid floor, and forms an α-helix in triplexes except for Tc (disordered) (Fig. 4e-g). Compared with published N-terminal regions of Tri1 of other herpesviruses

structures[18,19,21–23,25], we note that the long Tri1 N-terminal anchored domains of HSV and VZV are disordered. More ordered densities are found for β- and γ-herpesviruses, which bear shorter Tri1 N-terminal anchored domains (i.e., they always form helices) (Fig. 4h and Supplementary Fig. 11a).

In addition to anchored interactions under the capsid floor, the three subunits of the triplex also connect with each other at the buttress domain (aa 1129–1136) of the neighboring hexon MCP (Supplementary Fig. 12). Such interactions between the triplex and MCP render the triplex a hub, facilitating connection of the MCPs and further stabilizing the PRV capsid structure.

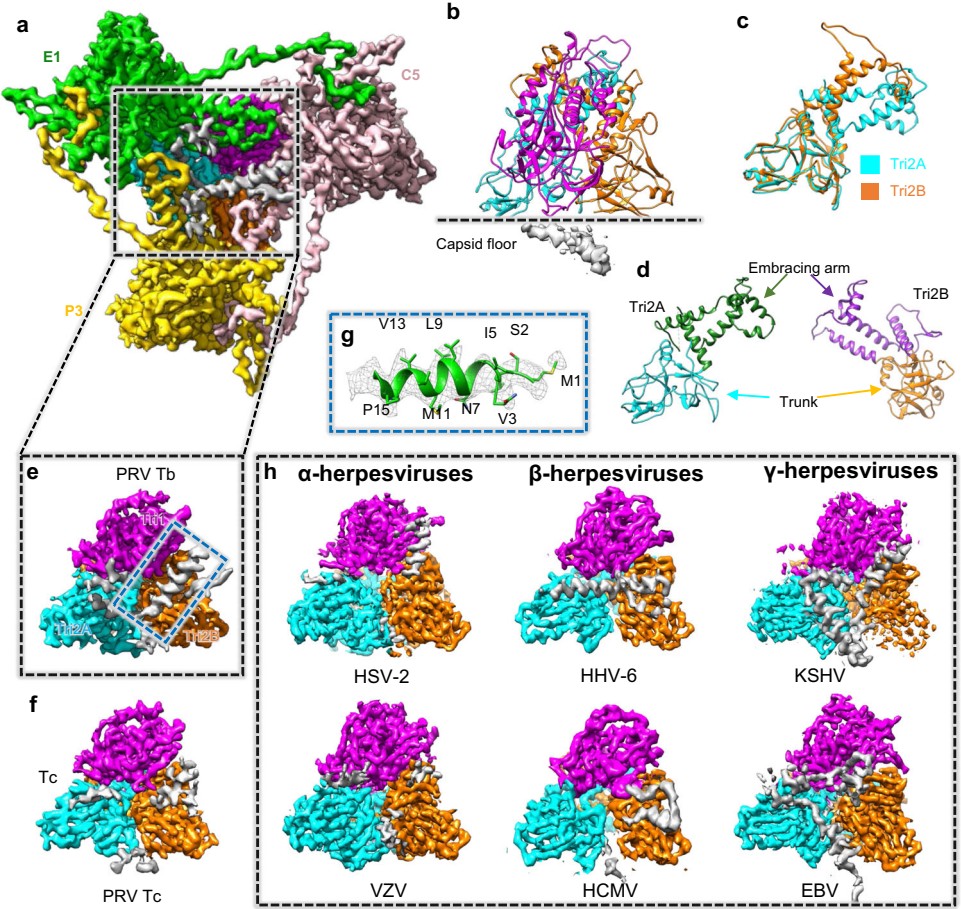

**Fig. 4 Structure of the triplex and characterization of the Tri1 N-terminus. a** Inside-out view of the density map of triplex Tb with surrounding MCPs. **b–d** Atomic model of Tb (**b**) and its components Tri2A and Tri2B (**c**, **d**). Superposition of Tri2A and Tri2B shows consistency in their trunk domains and conformational differences in their embracing arm domains (**c**). **e**, **f** Density maps of triplex Tb (**e**) and Tc (**f**) shows distinctions in the under-floor densities (colored in gray). **g** A close-up view of the density map (gray mesh) of the Tri1 N-terminal region with the fitted atomic model (ribbon/sticks) shown in the dashed box. **h** Comparison of the under-floor densities of Tbs from different herpesviruses. Abbreviations: MCP Major capsid protein.

**Structure of the SCP and its interaction with MCPs.** The small capsid protein (SCP) of PRV is encoded by UL35 and has only 103 amino acids. This makes the PRV SCP the shortest—albeit the most diverse in sequence—of all capsid proteins in the α-herpesvirus family (Supplementary Fig. 11b and Supplementary Table 2). Nevertheless, PRV SCP is structurally conserved and highly similar to HSV and VZV[17,19] (Fig. 5a). PRV SCP caps only hexon but not penton (Fig. 5b), which is a distinctive feature of those α-herpesviruses. Structurally, the PRV SCP consists mainly of two parts: a helix-rich domain in its N-terminus (aa 1-68) that is connected to a long insertion hairpin-loop in its C-terminus (aa 69-103) (Fig. 5a). The former forms a classic helix-turn-helix and interacts with MCP, whereas the latter inserts into the pocket formed by adjacent MCPs and therefore stabilizes SCP-MCP binding (Fig. 5c). Amino acids at positions 32 to 41 of SCP are highly flexible and could not be built in the atomic model (Fig. 5a).

Six SCPs, corresponding to six MCPs in one hexon, connect each other end-to-end to form a hexagram cap-like structure that covers the hexon (Fig. 5b). The hairpin-loop domain in each SCP plays an important role in mutually connecting SCPs and MCPs, which are directly inserted into the gap formed by the two adjacent MCPs, with hydrophobic interactions between MCP and SCP forcing a close interaction. At the same time, the N-terminus of the adjacent SCP also interacts with the hairpin-loop, further strengthening the stability of the hexagrams (Fig. 5b-c). In addition, electrostatic analysis shows that a positively charged stretch of residues 65-75 of SCP electrostatically interact with the negatively charged MCP upper region (aa 820-828) (Fig. 5d). The interactions we observed between PRV SCPs and SCP-MCP are very similar to those of VZV reported previously[19]; although, VZV SCP is much larger (235 residues). These non-covalent interactions enable SCP to further stabilize the structure of the hexamer MCP and thus stabilize the viral capsid.

**Structures of the DNA-translocating portal and CATC.** To better define the structure of the portal complex and attain a structural basis for genome packaging and organization in the PRV C-capsid, we performed a 3D classification using the 5-fold sub-particles of the C-capsid. A unique class representing portal vertex contained 8.19% (~1/12) were classified as portal vertex sub-particles (Supplementary Fig. 13). 3D refinement-imposed 5-fold symmetry was then performed, and the density map of the 5-fold sub-particles of the C-capsid containing the portal complex was obtained at 6.8-Å resolution (Fig. 6d and Supplementary Fig. 13). Based on the orientations of the portal vertex, we further reconstructed the portal based C5 symmetry capsid structure to a resolution of 4.7 Å (Fig. 6a-b, Supplementary Fig. 13 and Supplementary Movie 1). The portal-containing capsid reconstruction revealed an ordered, layered dsDNA around a less-ordered core (Fig. 6a-b and Supplementary Movie 1). At least six recognizable layers of dsDNA densities were observable (Fig. 6b).

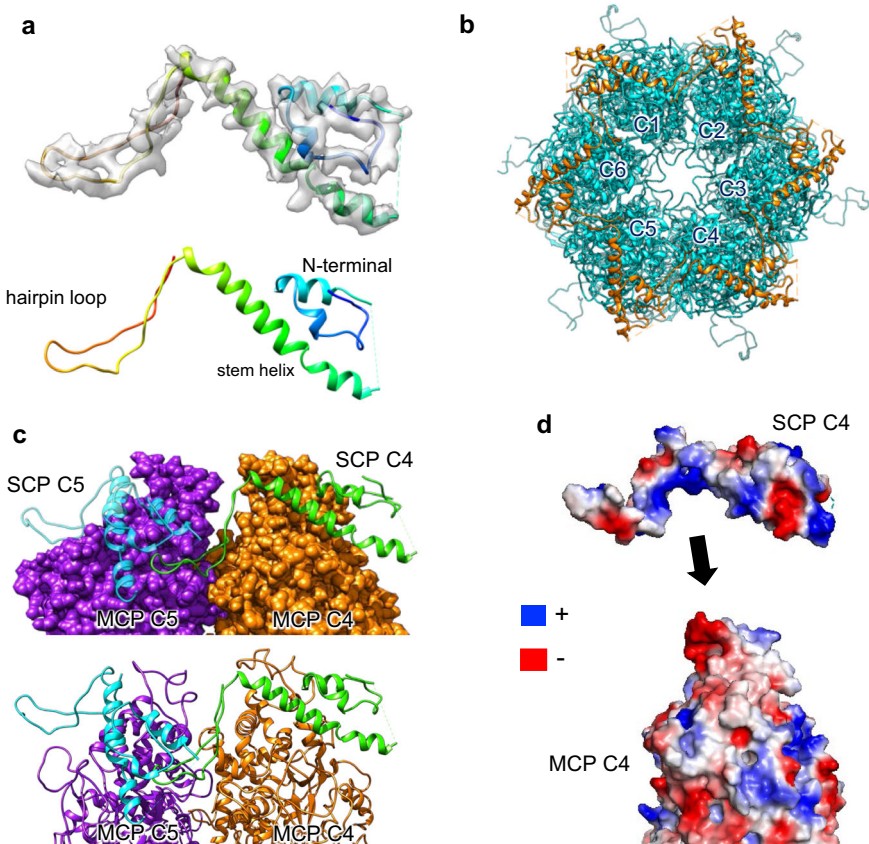

**Fig. 5 Structure of SCPs and interactions with MCPs. a** Atomic model of an SCP monomer shown as ribbons, rainbow-colored from the N-terminus (blue) to the C-terminus (red), with (up) and without (low) the corresponding density map (gray surface). **b** Crosslinking of six SCP subunits to form a gear-shaped hexameric ring crowning the C hexon. **c** SCP bridging two underlying MCPs (MCP C4 and C5). The hairpin loop of SCP C4 inserts into a groove formed by MCP C5 and SCP C5. **d** Open-book view of the electrostatic potential surfaces of SCP C4 and MCP C4 showing complementary electrostatic surface charges. Abbreviations: MCP Major capsid protein; SCP Small capsid protein.

The overall structure of the portal complex of PRV resembles that of HSV-1[27], KSHV[26], and EBV[24,25], comprising a 12-fold symmetrical portal (C12 portal) in the lower layer and 5-fold symmetrical tentacle helices in the upper layer (Fig. 6c). The C12 portal structure consists of 12 identical pUL6 monomers surrounded by a circular DNA anchor. Using this medium-resolution density map, we could well fit the previously reported portal structure of HSV-1[27] (Fig. 6f and Supplementary Fig. 14).

The 5-fold symmetrical tentacle helices, which was previously postulated to belong to the unmodeled residues of the portal clip of pUL6[27], was suggested to bind terminase to mediate the package of viral genome[36]. The portal vertex reconstructed at 4.7 Å resolution under C5 symmetry clearly demonstrated visible Cα bumps and allowed us to build the Cα skeleton model of the tentacle helices, and identified the formation of a scaffold structure composed of five groups of α-helixes, with each group containing a long and a short coiled-coil α-helix. We showed that the terminal DNA penetrated the central channels of both the C12 portal proteins and C5 tentacle helices (Fig. 6c, f).

A comparison of the penton vertex and the portal vertex of the PRV C-capsid shows nearly identical arrangement of the surrounding hexons (P-hexons) (Fig. 6d-e). Five CATCs form a star-shaped tegumental layer that is situated on both the penton and portal vertices, forming extensive interactions with the penton and portal proteins to stabilize the conformation of the vertex complexes (Fig. 6d-e). Like other herpesviruses, each CATC of PRV bridges the space between the penton/portal and hexon by binding at two triplexes, Ta and Tc (Fig. 6f). Triplex Ta

also interacts via its trunk domain with the tentacle-like helix (C5 portal) (Fig. 6f). These interaction networks mediated by portal proteins, CATCs and triplexes stabilize the entire portal complex.

Previous studies showed that the CATC structure of PRV is a hetero-pentamer composed of one pUL17 protein, two pUL25 proteins, and two pUL36 proteins[30]. Based on our localized reconstruction of the C-capsid 5-fold sub-particle, an atomic model of PRV CATC was built; however, only the N-terminus of the pUL25 subunits (upper pUL25 aa 1-73; lower pUL25 aa10-83) could be built out as α-helices; the remaining C-terminal globular domain, in contrast, was disordered and failed to be modeled (Fig. 6g). Nonetheless, density representing the C-terminal head of pUL25 could be observed in the unsharpened density maps of both penton and portal vertexes, which may contribute to the penton and portal cap (Fig. 6h-i and Supplementary Fig. 14).

A comparison of the penton and portal vertexes also revealed a slight difference (~6°) in the orientations of the bound CATCs (Fig. 6j-k); such discrepancies may influence their interactions with the triplexes underneath. Indeed, we observed that the peri-portal Triplex Ta rotates counterclockwise by ~120° when compared to peri-penton Ta (Fig. 6l). Similar observation was found in other α-herpesviruses such as HSV-1[27] and HSV-2[37], as well as γ-herpesviruses EBV[24,25] and KSHV[26].

## Discussion

There are technical challenges in determining the structures of large particles like the herpesvirus. One such challenge is the enormous

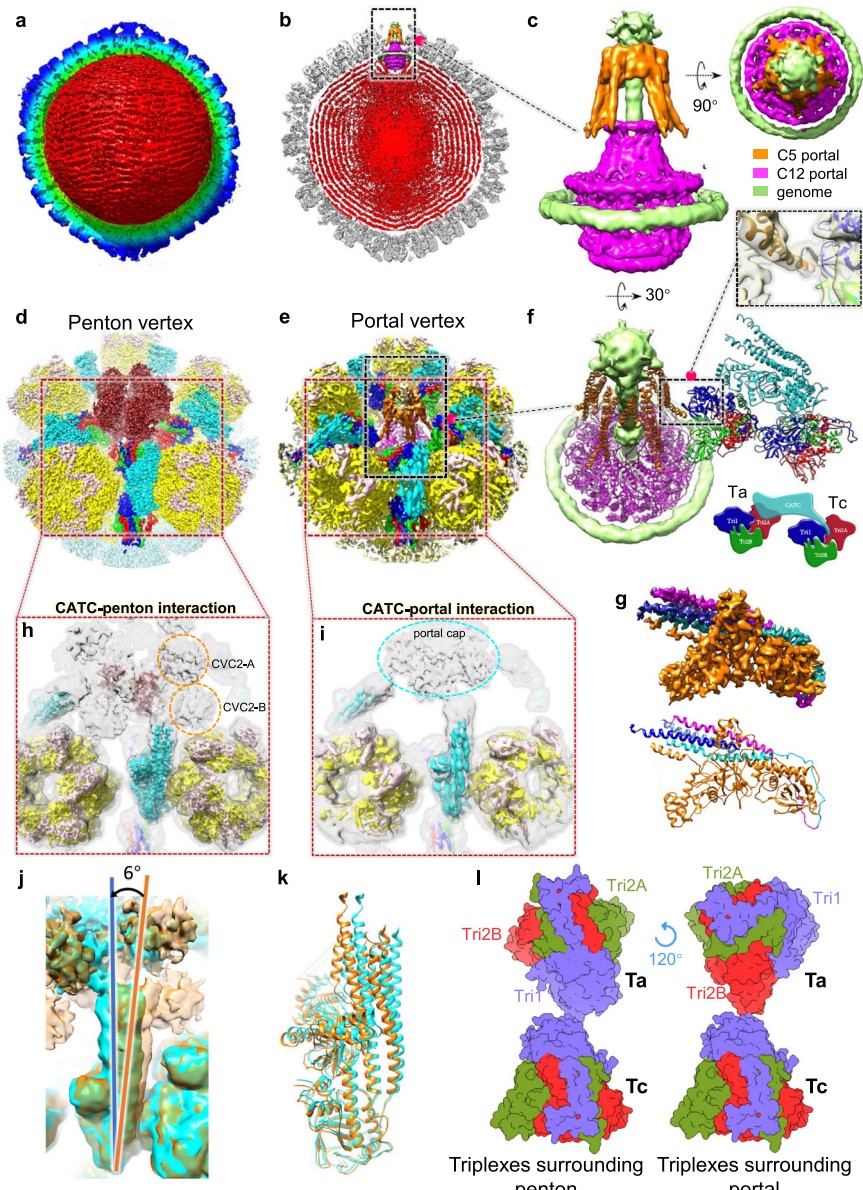

**Fig. 6 Architecture of the PRV C-capsid with portal and CATCs. a** The unsharpened C5 reconstructed maps of the PRV C-capsid is presented showing a cropped view of the interior density. **b, c** Clipped (**b**) and zoomed-in (**c**) views of the C5 capsid reconstruction showing packaged double-stranded (ds) DNA within the capsid (**b**) and structural components around the portal vertex (**c**). **d, e** Sub-particle reconstructions of the penton vertex (**d**) and the portal vertex (**e**). **f** Rotated view of (**c**) with a fitted model of the C5 portal protein (helix bundle) and the C12 portal protein. This view shows the interactions between the portal proteins and CATCs and triplexes beneath. **g** Segmented density map of the CATCs and the corresponding model. **h** Penton vertex sub-particle reconstruction. The CATC helix bundle is near two densities (orange circles), which are likely the head domains of capsid vertex component 2 (CVC2)-A and CVC2-B. **i** Portal vertex reconstruction. The CATC helix bundle is connected to the portal cap (cyan circle). **j, k** Comparisons of maps (**j**) and corresponding models (**k**) of CATCs from the penton and portal vertices. There is a counterclockwise (6°) rotation of the CATCs when bound to the portal vertex. **l** Comparison of triplexes: Ta in external orientation relative to Tc in the penton and portal vertexes. Ta in the penton vertex rotates 120° counterclockwise around its apical domain. Abbreviations: CATC Capsid-associated tegument complex; MCP Major capsid protein; SCP Small capsid protein; Tri Triplex.

particle diameter (diameters of ~200 nm for virion and ~128 nm for capsid) that creates a focus gradient across a large sample thickness, violating the Central Projection Theorem with the introduction of a curved Ewald sphere[38]. The efforts of many investigators over decades have pushed the structural resolution of herpesvirus capsids to near-atomic levels through cryo-EM[17–25]. Two major aspects have led to the high-resolution reconstructions of herpesviruses: the first, is data collocation via state-of-the-art Titan Krios TEM and K2/K3 cameras; the other, lies in the usage of some advanced

reconstruction strategies such as "block-based" reconstruction[39]. Here, we sought to reconstruct the near-atomic PRV capsid structure using a TF30 electron microscope (TF30) equipped with a Falcon 3 detector. Although without the use of a highly stable illuminate system or a more stringent parallel beam condition, and even under linear mode conditions without electron counting for data collection, we materialized the high-resolution reconstructions of the herpesvirus A- and C-capsids by the meticulous combination of localized reconstruction.

Three distinct types of herpesvirus capsids have been detected within the virus life cycle: A-, B-, and C-capsids. The A-capsid is empty, whereas the C-capsid is fully packed with genomic DNA, and the B-capsid contains a core of scaffold proteins but without genomic DNA. It has been speculated that A- and B-capsids assemble with abortive genomic DNA packing[18]. In this study, we found the C-capsid is the dominant component while only a small fraction of A-capsids and few B-capsids in the samples purified from supernatant of PRV-infected cells (Supplementary Fig. 2), so we just resolved the structures of A- and C-capsid but not B-capsid. The structures of the PRV A-capsid and C-capsid are nearly identical except for the presence of CATCs in the latter. The overall structure of the PRV capsid is highly similar to that of other α-herpesviruses[17–20], especially to that of VZV, which similarly harbors looser penton-hexon interactions and triplex-MCP interactions[19,20]. We previously demonstrated that The VZV capsid has comparable high thermal stability, although with relatively fewer intra- and inter-capsid protein interactions and less stably associated tegument proteins compared with other human herpesviruses such as HSV, probably due to its relatively smaller size of genome (~125 kb versus ~152 kb for HSV)[35]. However, in this study, the thermal stability of PRV was revealed significant weaker than that of HSV-1 (Supplementary Fig. 10). The genome size of PRV (about 143 kb) was greater than VZV (about 125 kb) but more similar to HSV (about 152 kb). We speculated that the capsid of PRV may less stable to that of HSV-1 and VZV due to its weaker intra- and inter-subunit interactions (compared to HSV-1) and larger genome size (compared to VZV).

The overall structure of PRV portal vertex is similar with those revealed in other herpesviruses, indicating herpesviruses may follow the same modality for genome organization and releasing. The current structural studies including PRV showed that the peri-portal Triplex Ta always rotate counterclockwise by ~120° when compared to peri-penton Ta[25–27,37]. However, as for γ-herpesviruses such as KSHV[26] and EBV[25], the peri-penton Ta was found also rotating ~120° when CATC bound. In contrast, our structural information confirmed that CATC binding or not does not affect Ta orientation at the penton vertex, as the orientations of Ta from A-capsid (CATC absent) or C-capsid (CATC present) are the same. Studies on HSV-1[27] and HSV-2[37] also confirmed that binding of CATCs at penton vertex does affect the Ta orientations, which should be a distinctive structural feature between α- and γ-herpesviruses. In addition, PRV capsids also show some minor structural features reminiscent of β- or γ-herpesviruses. For example, the short N-terminal anchor sequence of PRV triplex Tri1, which forms an α-helix, is similar to that of HHV-6[22].

By carrying out 3D classifications and localized reconstructions of the 5-fold sub-particles, we also successfully obtained a moderate-resolution structure of the portal vertex of the PRV C-capsid, revealing its similar structural architecture to that of other herpesviruses[24–27]. However, the same strategy was unable to procure a 3D reconstruction of the portal vertex of the PRV A-capsid, possibly indicating the instability or altered structural dynamics of the portal vertex in the A-capsid. Further efforts are needed for the classification of the A-capsid portal structure as well as the high-resolution reconstructions of both the A- and C-capsid portal vertices.

Herpesviruses are potential oncolytic viruses and, in recent decades, have been recognized as cutting-edge biological therapies for cancer treatment. For instance, an HSV-1-based oncolytic virus, T-VEC, has been approved for the treatment of melanoma[40,41]. Although PRV may have potential risk in human disease, compared with HSV-1, PRV is a less harmful option, and may thus offer an advantage as a potential oncolytic virus. Indeed, a few previous studies have shown PRV to be an oncolytic virus that can replicate effectively in cancer cells and lead to their rupture[42–44]. PRV has a higher safety profile in primates than other mammals, and most humans have no pre-existing neutralizing antibodies against PRV that may inhibit the virus from targeting tumor tissues. PRV may therefore be an ideal gene therapy vector. Yet, despite its potential advantages, PRV will likely only be able to target specific types of cancer cell lines. Thus, before its utility as an oncolytic agent, PRV may require further modification if it is to be used to target a larger range of tumors types[45]. Considering that PRV has a moderate-sized genome compared with other herpesviruses, along with a relatively loose layer spacing, as shown in this study, PRV should tolerate modification to its genome (e.g., insertion) to some extent. Furthermore, although a presumed higher safety profile compared with other oncolytic viruses, the potential hazards associated with PRV will need to be considered. Indeed, several cases of human viral encephalitis caused by PRV infection have been reported, with patients presenting with respiratory dysfunction and acute neurological symptoms[46]. Thus, before PRV is more readily identified in humans as a cause of disease, there is an urgent need to construct PRV oncolytic strains to take advantage of its presumed higher safety and efficacy against tumors.

To date, there are no approved drugs to treat the disease caused by PRV infection. Consequently, there is also an urgent need to develop anti-PRV therapeutics for potential pandemics in swine and against viral transmission in humans. We also need to control the potential risk of PRV in oncolytic applications. Traditional drugs against herpesvirus have limited inhibitory activities against PRV, and more and more resistant herpesvirus strains have been reported[47]. Thus, it is necessary to develop a broad-spectrum antiviral drug against a variety of herpesviruses. Here we point to CATCs, consisting of pUL25, pUL28, and pUL33, as having an important function in viral particle assembly, with pUL25 highly conserved in herpesviruses (Supplementary Table 2). Indeed, others have shown that deletion of UL25 can significantly decrease the efficacy of PRV replication[48]. We resolved the near-atomic structure of the PRV CATC and confirmed structural similarities among CATCs from α- and γ-herpesviruses. Thus, UL25 could provide a promising target for the design of a universal antiviral therapeutic. Furthermore, antiviral drugs based on the CATCs themselves may be a reasonable way to create a broad-spectrum therapeutic against herpesviruses.

Overall, the near-atomic structures of the PRV A- and C-capsid reported here reveal the various structural similarities and differences among α-herpesviruses and provide comprehensive insight into the mechanism of capsid assembly. Furthermore, the portal complex of PRV is shown to be decorated by CATCs. We show the overall structure of the PRV portal vertex to be highly similar to that of HSV-1, but with some distinctions common only to γ-herpesviruses. These results provide a glimpse of the structural uniqueness of PRV capsids and shed light on how to capitalize on this knowledge for the design of PRV-based therapeutics.

## Methods

**Cells.** PK-15 (ATCC CCL-33) and Panc-1 (ATCC CRL-1469) cells were purchased from the American Type Culture Collection and cultured in Dulbecco's modified Eagle's medium (DMEM) (Invitrogen) supplemented with 10% heat-inactivated fetal bovine serum (FBS) (Gibco), 100 μg/mL streptomycin and 100 IU/mL penicillin at 37 °C in a humidified 5% $CO_2$ atmosphere. Mycoplasma contamination of cells was tested as negative by PCR before experimentation.

**Viral capsids preparation.** PK-15 cells were infected with the Bartha strain of PRV at a MOI of 0.01. At 3 days post infection (dpi), culture medium was collected and centrifuged under 1500 g at 4 °C for 10 min to remove large cell debris. The supernatant was collected and then precipitated by PEG8000 (Apollo Scientific Ltd) (final concentration of 8% PEG8000). The pellet was resuspended in phosphate buffered saline (10 mM PBS, pH 7.4) and further purified by centrifugation through a linear 20% to 50% (w/v in PBS) sucrose gradient at 80,000 g at 4 °C for 4 h. The light-scattering band of viral particles was collected and dialyzed with PBS,

then centrifuged through a 20% (w/v in PBS) sucrose cushion at 80,000 g at 4 °C for another 4 h. The pelleted crude PRV viral particles were finally resuspended in PBS.

**Transmission electron microscopy.** Panc-1 cells infected with PRV Bartha strain (at a MOI of 1) were collected at 24 h post infection and prepared for transmission electron microscopy (TEM) analysis, as previously described[49]. In brief, samples were fixed in 2.5% glutaraldehyde in phosphate buffer (0.1 M, pH 7.4), post-fixed with 1% osmium tetroxide, incubated in 1% aqueous uranyl acetate overnight, dehydrated through graded ethanol, and then embedded with an Embed 812 kit (Electron Microscopy Sciences). Ultra-thin sections were stained with 3.5% aqueous uranyl acetate and 0.2% lead citrate. Images were recorded using a Tecnai G2 Spirit transmission electron microscope (FEI).

**Analytical ultracentrifugation.** Sedimentation velocity experiments were performed on a Beckman XL-A analytical ultracentrifuge at 20 °C. Purified PRV capsids were diluted with PBS buffer to 400 μL and with A280 nm absorption of about 0.8. Samples were loaded into a conventional double-sector quartz cell and mounted in a Beckman four-hole An-60Ti rotor. Data were collected at a wavelength of 280 nm during 655 g centrifugation. Interference sedimentation coefficient distributions were calculated from the sedimentation velocity data using the SEDFIT software program (www.analyticalultracentrifugation.com).

**Particle-stability thermal-release assay.** The viral particle stability thermal-release assay was performed in 96-well PCR plates by use of a CFX86 PCR instrument (Bio-rad) as previously described[35]. Each 50 μL reaction was set up in a thin-walled PCR plate containing 1.0 μg of purified capsids of PRV or HSV-1 and 5 μM SYTO9 green-fluorescent nucleic acid stains (Thermo Fisher Scientific) in PBS. The temperature was ramped from 25 to 99 °C and the fluorescence was recorded in triplicate at 0.5 °C intervals. The negative first derivatives of the fluorescence change ($-d(RFU)/dT$, where RFU is relative fluorescence units and T represents time) were plotted against temperature.

**Cryo-EM imaging.** A 3 μL aliquot of purified PRV samples was applied to freshly glow-discharged holey carbon Quantifoil Cu grids (R2/2, 200 mesh, Quantifoil Micro Tools), and then blotted for 6 s before plunge-freezing the grids into liquid ethane cooled by liquid nitrogen inside a Vitrobot Mark IV (Thermo Fisher Scientific) at 100% humidity and 4 °C. Cryo-EM images of the PRV capsids were acquired with the FEI Tecnai F30 TEM (Thermo Fisher Scientific) with a Falcon 3 direct electron detector (Thermo Fisher Scientific) at a nominal 93,000× magnification, corresponding to a calibrated physical pixel size of 1.117 Å. Each movie contained 39 frames, with a total dose of 30 e$^-$/Å$^2$ at an exposure time of 1 s. Data were automatically collected with Thermo Fisher EPU software.

**Image processing and 3D reconstruction.** Drift and beam-induced motion correction were performed with MotionCor2[50] to produce a micrograph from each movie. Contrast transfer function (CTF) fitting and phase-shift estimation were conducted with Gctf[51]. Micrographs with astigmatism, obvious drift, or contamination were discarded before reconstruction. Particles were automatically picked and screened using cisTEM[52]. Several rounds of reference-free 2D classifications and unsupervised 3D classifications were executed using Relion 3.0[53]. Sorted particles were then subjected to final homogenous refinement using Relion 3.0 and cisTEM. The resolution of all density maps was determined by the gold-standard Fourier shell correlation curve, with a cutoff of 0.143[54]. Local map resolution was estimated with ResMap[55].

To improve the resolution, we used Relion symmetry expansion to extract and perform sub-particle reconstruction, as previously described[56]. Briefly, after 3D refinement with imposition of icosahedral symmetry, we extracted sub-particles from the 2-fold, 3-fold, and 5-fold regions, respectively, in a box-size of 400 to 480 pixels and corrected the defocus value of the sub-particles by Scipion[57]. The extracted sub-particles were used to generate the initial model using the Relion *relion_reconstruct* command. Further 3D classification and refinement were performed by relion 3.0 or cisTEM. The resolution was assessed by Fourier shell correlation curves with a cutoff at 0.143 from two independent half-sets of the sub-particles.

**Atomic-model building, refinement, and 3D visualization.** The initial PRV capsid model was generated from homology modeling based on the atomic model of HSV-1[58] by Accelrys Discovery Studio software (available from: URL: https://www.3dsbiovia.com). We initially fitted the templates into the corresponding segmented volume (enclosing an asymmetric unit) of the final cryo-EM maps using Chimera[59], and further corrected and adjusted them manually by real-space refinement in Coot[60]. The resulting models were then refined with *phenix.real_space_refine* in PHENIX[61]. These operations were executed iteratively until the problematic regions, Ramachandran outliers, and poor rotamers were either eliminated or moved to favored regions. After several cycles of refinement, the resulting models were fitted into the map with six neighboring asymmetric units. The totality of these seven asymmetric units were subjected to further real-space refinement to optimize clashes. The final atomic models were validated with

Molprobity[62,63]. All figures were generated with Chimera or Pymol software (http://www.pymol.org).

**Reporting summary.** Further information on research design is available in the Nature Research Reporting Summary linked to this article.

## Data availability
The data that support this study are available from the corresponding authors upon reasonable request. Structure coordinates generated in this study have been deposited in the Protein Data Bank under accession codes 7FJ3 (A-capsid) and 7FJ1 (C-capsid). The corresponding EM density maps have been deposited in the Protein Data Bank under accession numbers EMD-31612 (icosahedral reconstruction of the A-capsid), EMD-31611 (icosahedral reconstruction of the C-capsid), EMD-31610 (portal vertex), EMD-31616 (portal based reconstruction of the C-capsid), EMD-31593 (2-fold sub-particle reconstruction of A-capsid), EMD-31592 (3-fold sub-particle reconstruction of A-capsid), EMD-31591 (5-fold sub-particle reconstruction of A-capsid), EMD-31609 (2-fold sub-particle reconstruction of C-capsid), EMD-31608 (3-fold sub-particle reconstruction of C-capsid) and EMD-31594 (5-fold sub-particle reconstruction of C-capsid).

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

## Acknowledgements

This work was supported by grants from the National Natural Science Foundation (grant no. 82073756 to Y.C.), the Science and Technology Major Projects of Xiamen (grant no. 3502Z20203023 to Q.Z.).

## Author contributions

Q.Z., Y.C., S.L., and N.X. designed the study; G.W., P.H., T.C., and L.L. prepared the virus sample; Z.Z., H.S., M.H., and Z.C. prepared the cryo-EM grids and record the cryo-EM movies; Q.Z., Z.Z., Y.H., H.S., M.H., T.L., and H.Y., processed data, obtained all 3D reconstructions; Z.Z., G.W., Y.Q., Z.K., Y.G., J.Z., S.L. Q.Z., Y.C., and N.X. analyzed data. Z.Z., G.W., Q.Z., and Y.C. wrote the manuscript. Z.Z., G.W., Q.Z., S.L., Y.C., and N.X. participated in discussion and interpretation of the results. All authors reviewed and approved the paper.

## Competing interests

The authors declare no competing interests.
