## [Peer Review File · Nature Communications]

Structures of Pseudorabies Virus CapsidsREVIEWER COMMENTS

Reviewer #1 (Remarks to the Author):

In this manuscript, the authors present a comprehensive structure description of the pseudorabies virus (PRV) capsid at near-atomic resolution with advanced cryoEM techniques such as localized refinement and classification. Their structures also include the capsid-associated tegument complex and the unique portal vertex, structures of which are technically very challenging to solve. Although most of the structures described here are so similar to those of other herpesviruses, particularly other alpha-herpesviruses, that there was no gratifying surprises when reading this manuscript, I still fully support the publication of this paper because I believe it will become a very valuable resource for many veterinary researchers interested in PRV virology and benefit many potential PRV-based applications as discussed extensively in this manuscript. However, a major mistake and several minor issues should be corrected before it can be accepted for publication, as detailed below.

Major revision:

There is a major mistake in the result interpretation. In lines 268-273 of the Result section, and lines 305-308 of the Discussion, the authors described that the triplex Ta surrounding the portal vertex is rotated 120 degree counterclockwise compared to Ta surrounding penton vertexes. This structural description is correct. However, the authors claim that such rotation of Ta at the portal vertex was only observed in gamma-herpesviruses KSHV and EBV, but not in alpha-herpesvirus HSV-1 (this is not correct). Therefore, they claim it as a major structural difference between PRV and other alpha-herpesviruses, in addition to its short N-terminal anchor in the Tri1 protein that is more reminiscent of beta- and gamma-herpesviruses. This represents a misunderstanding of the other herpesvirus structure papers.

In fact, in all herpesvirus capsids with the portal vertex structure determined, it has been observed that the triplex Ta surrounding the portal is always rotated 120 degree compared to the penton vertex Ta. This rotation of Ta seems to be accommodating the interactions between triplex Ta and the portal. In reference 32 cited by the authors to support their claim, the relative rotation of portal vertex Ta was actually clearly depicted in Fig.4 e and f.

The true difference between gamma-herpesviruses and alpha-herpesviruses regarding this issue is that: in gamma-herpesvirus penton vertices, Ta is also rotated (similar to those at the portal vertex) when there is CATC binding; while in alpha-herpesviruses, CATC binding or not does not affect Ta orientation at the penton vertex. This logic also suggests that there is no causative relationship between CATC binding and Ta rotation at the portal vertex in alphaherpesviruses, as the authors imply in line 271 ("binding of CATCs to the portal vertex led to counterclockwise rotation of the triplex Ta by 120°").

Suggested minor revisions:

The first two paragraphs of the introduction can be more concise. The current version reads like a lengthy review and contains too much detail not relevant to the structure study.

Line 36, the disease caused by PRV is named "Aujeszky's disease", to be scientifically more accurate.

Line 82, the authors used "the middle part of the hexamers and pentamers" to describe positioning of the triplexes. This is inaccurate and confusing.

Line 89, "A-capsids" and "C-capsids" warrant an explanation for readers not familiar with the jargon.

Line 152, "portal floor" should be "capsid floor".

Line 155, "22 Å" should be specified as "22 Å in diameter".

Line 244, the enzyme that packages the genome in herpesvirus is named "terminase".

Line 372-381 the method for sample preparation is confusing and seems to be inaccurate, and needs clarification. For example, "repeated freeze-thawing three times" and "the resulting solution was centrifuged through a linear sucrose gradient" suggest that the described protocol was for purification of PRV capsids from pelleted cells. But the authors also mentioned "the culture medium were collected", and the final product used for cryoEM was described as "the pelleted crude PRV virions". If culture supernatant was also used to purify intact PRV virions, and combined with PRV capsids from inside the cells, it should be clearly stated and I imagine there should be at least one step of pelleting the virions from the supernatant before it was centrifuged through the density gradient.

Line 412, "exact" should be "extract".

Reviewed by: Xinghong Dai

Reviewer #2 (Remarks to the Author):

Comments:

The manuscript titled "Atomic Structures of Pseudorabies Virus Capsids" by Guosong Wang et al resolved the capsid structures of PRV A and C particles by cryo-EM. By using sub-particle reconstruction strategy, they obtained near atomic structures of both capsids. Then the authors analyzed the structures of MCP and inter/intra interactions between MCPs. The author also further analyzed the structures of triplex, SCP, portal of C-capsid and CATC. Overall, the experimental work is of high quality and it potentially deserves the publication in Nat Commun. However, a number of concerns have to be addressed before the acceptance.

Major points:

1. The authors claimed that they reported the near-atomic structures of the PRV A-capsid and C-capsid, and illustrate the interaction that occurs between these subunits. While it seems not convincing to define A-capsid or B-capsid by electron microscopy observation. The authors should determine the sedimentation coefficient of these particles by analytical ultracentrifugation to exactly verify A-capsid or B-capsid.
2. Lines 168-175. Compared to HSVs, PRV MCPs exhibit much less inter-subunit interactions between the penton MCP and the adjacent hexon due to the loose and disordered N-lasso and dimerization domains of the penton MCP. Are HSV capsids more stable than PRV capsids? It's necessary to provide functional insights/implications from structural details. Or perhaps correlated with the small-sized genome for PRV and VZV, when compared to HSV and HCMV.
3. Lines 194-198. Such interactions between the triplex and MCP not only determine the orientation of the triplex, but also render the triplex a hub, facilitating connection of the MCPs and further stabilization of the PRV capsid structure. It seems not right. It's not clear/reasonable that the interactions between the triplex and MCP determine the orientation of the triplex. The triplex lies among three adjacent capsomers, surrounded by at least three buttress domains of three neighbouring MCPs. HOW and WHY such interactions determine the orientation of the triplex??
4. The portal structure in herpesviruses consists of 12 identical pUL6 monomers. But it's very confused why the 12-fold symmetrical portal possesses 5-fold symmetrical tentacle helices in the upper layer? It's weird that the 12 identical pUL6 monomers adopt a 5-fold symmetrical structure, where are the left two? The authors must clarify this point clearly. What evidences were used to support that the 5-fold symmetrical tentacle helices belong to the portal? Why not other protein components?
5. It has been reported that N-lasso and dimerization domain of EBV and HSV Hexon MCPs have various configurations, however, only one conformation of the Hexon MCPs was found in PRV. Do all these MCPs have the same conformation?

Minor points:

1. Lines 103-104. Many PRV virions as well as naked capsids had been secreted into the culture supernatant. It might not be accuracy. Generally, mature PRV virions with outer membrane are secreted, but naked capsids are not. More probably, the observed naked capsids in supernatant were produced due to the mechanical/physical forces during the purification. Did the authors check the cell supernatant prior to any purification, including concentration?
2. Lines 137-139. The models of the asymmetric units of the A- and C-capsids are nearly identical, with a root mean square deviation of 0.298 Å over 26,091 atoms, with the C α positions of the two models aligned (Fig. S6). In general, it should not be like this. Due to the huge pressure produced by genome

packaging in C-capsids, at least the portal structures differ radically, including the morphology and location in the capsids (Lokareddy, R.K. et al., Nat Commun 2017; Wang Nan., et.al, Yang Yunxiang.et.al, Protein & Cell 2020).

3. Line 142. 150-kD should be 150-kDa.

4. In Fig3g, only two β -sheets can be seen instead of the five β -sheets described in the text.

5. Does buttres of MCP hexon and penton have different conformation?

6. In Fig6c, the color bar of 'genome' should be changed to green instead of blue.

7. Does the Fig3e represent the MCP-MCP type II interaction? These are totally inconsistent with the descriptions in line 161-163.

8. Why the-terminal anchored domain of triplex Tc is disorder?

9. The rotation of the triplex Ta adjacent to portal has been reported in HSV-2 and other herpesviruses (Wang Nan et.al Protein & Cell 2020; Hong Zhou. et.al, Cell 2020). It may be universal organization for the assembly of herpesviruses.

10. Lines 244-245. The 5-fold symmetrical tentacle-like helix (C5 portal) is supposed to bind terminal enzymes to mediate packaging of the genome. The terminase complex in herpesviruses was reported to form a hexamer (revealed by cryo-EM structure), not a pentamer.

Response to Reviewer Comments on the Manuscript NCOMMS-21-28568

We thank the two reviewers for recognizing the merit of our work and for suggesting ways to improve our manuscript. As you will see in the following itemized responses, we have addressed all the concerns with appropriate additional experiments and analyses, and revised our paper accordingly. To facilitate the navigation of this document, we copied the reviewers' comments verbatim in **blue** and typed our responses in **black**.

Reviewer #1

General opinion and comments:

Reviewer: In this manuscript, the authors present a comprehensive structure description of the pseudorabies virus (PRV) capsid at near-atomic resolution with advanced cryoEM techniques such as localized refinement and classification. Their structures also include the capsid-associated tegument complex and the unique portal vertex, structures of which are technically very challenging to solve. Although most of the structures described here are so similar to those of other herpesviruses, particularly other alpha-herpesviruses, that there was no gratifying surprises when reading this manuscript, I still fully support the publication of this paper because I believe it will become a very valuable resource for many veterinary researchers interested in PRV virology and benefit many potential PRV-based applications as discussed extensively in this manuscript. However, a major mistake and several minor issues should be corrected before it can be accepted for publication, as detailed below.

Major revision:

Comment 1: There is a major mistake in the result interpretation. In lines 268-273 of the Result section, and lines 305-308 of the Discussion, the authors described that the triplex Ta surrounding the portal vertex is rotated 120 degree counterclockwise compared to Ta surrounding penton vertexes. This structural description is correct. However, the authors claim that such rotation of Ta at the portal vertex was only observed in gamma-herpesviruses KSHV and EBV, but not in alpha-herpesvirus HSV-1 (this is not correct). Therefore, they claim it as a major structural difference between PRV and other alpha-herpesviruses, in addition to its short N-terminal anchor in the Tri1 protein that is more reminiscent of beta- and gamma-herpesviruses. This represents a misunderstanding of the other herpesvirus structure papers.

In fact, in all herpesvirus capsids with the portal vertex structure determined, it has been observed that the triplex Ta surrounding the portal is always rotated 120 degree compared to the penton vertex Ta. This rotation of Ta seems to be accommodating the interactions between triplex Ta and the portal. In reference 32 cited by the authors to support their claim, the relative rotation of portal vertex Ta was actually clearly depicted in Fig.4 e and f.

Response: We apologize for the negligence and thank the reviewer to bring this point out. We realized that the triplex Ta surrounding the portal vertex is indeed rotated 120 degree compared to the penton vertex Ta in HSV-1. This should be a common structural feature of the capsid of herpesvirus. We now revise the description in Result section, “*we observed that the peri-portal Triplex Ta rotates counterclockwise by $\sim 120^\circ$ when compared to peri-penton Ta (Fig. 6l). Similar observation was found in other α -herpesviruses such as HSV-1 and HSV-2, as well as γ -herpesviruses EBV and KSHV.*” (Page 14, lines 278-281)

Comment 2: The true difference between gamma-herpesviruses and alpha-herpesviruses regarding this issue is that: in gamma-herpesvirus penton vertices, Ta is also rotated (similar to those at the portal vertex) when there is CATC binding; while in alpha-herpesviruses, CATC binding or not does not affect Ta orientation at the penton vertex. This logic also suggests that there is no causative relationship between CATC binding and Ta rotation at the portal vertex in alphaherpesviruses, as the authors imply in line 271 (“binding of CATCs to the portal vertex led to

counterclockwise rotation of the triplex Ta by 120°”).

Response: Exactly right. For PRV, our structural information confirmed that CATC binding or not does not affect Ta orientation at the penton vertex, as the orientations of Ta from A-capsid (CATC absent) or C-capsid (CATC present) are the same. However, peri-portal Ta triplexes rotate ~120° counterclockwise related to peri-penton Ta. Given that the binding of CATCs on penton vertex does not affect Ta orientation, it’s arbitrary to say that binding of CATCs to the portal vertex led to counterclockwise rotation of the triplex Ta by 120°. Thus, we now tone down the causative relationship between CATCs binding and Ta rotation at the portal vertex and depicted this in Discussion section in details, and now reads: “*The overall structure of PRV portal vertex is similar with those revealed in other herpesviruses, indicating herpesviruses may follow the same modality for genome organization and releasing. The current structural studies including PRV showed that the peri-portal Triplex Ta always rotates counterclockwise by ~ 120° when compared to peri-penton Ta. However, as for γ -herpesviruses such as KSHV and EBV, the peri-penton Ta was found also rotating ~ 120° when CATC bound. In contrast, our structural information confirmed that CATC binding or not does not affect Ta orientation at the penton vertex, as the orientations of Ta from A-capsid (CATC absent) or C-capsid (CATC present) are the same. Studies on HSV-1 and HSV-2 also confirmed that binding of CATCs at penton vertex does affect the Ta orientations, which should be a distinctive structural feature between α - and γ -herpesviruses.*” (Pages 16-17, lines 321-331).

Suggested minor revisions:

Comment 3: The first two paragraphs of the introduction can be more concise. The current version reads like a lengthy review and contains too much detail not relevant to the structure study.

Response: As suggested, we have streamlined the first two paragraphs of the introduction and removed redundancies in the manuscript, please see the revised introduction at Page 3, lines 36-49.

Comment 4: Line 36, the disease caused by PRV is named “Aujeszky's disease”, to be scientifically more accurate.

Response: The related description has been revised.

Comment 5: Line 82, the authors used “the middle part of the hexamers and pentamers” to describe positioning of the triplexes. This is inaccurate and confusing.

Response: As suggested, we have rephrased the description as: “*The triplexes, each composed of one VP19C protein (Tri1, encoded by UL38) and two VP23 proteins (Tri2, encoded by UL18), anchor to the capsid floor via Tri1 N-anchor. Triplexes can be divided into six types (Ta, Tb, Tc, Td, Te, Tf) in terms of their localization on viral capsid.*” (Page 4, lines 69-72).

Comment 6: Line 89, “A-capsids” and “C-capsids” warrant an explanation for readers not familiar with the jargon.

Response: “A-capsids” and “C-capsids” were explained as “empty capsid resulted from abortive DNA packing (A-capsid)” and “mature capsids fully packed with DNA (C-capsid)” respectively, when they were mentioned for the first time in the manuscript. Please see Pages 4-5, lines 76-77 in the text.

Comment 7: Line 152, “portal floor” should be “capsid floor”.

Response: Corrected.

Comment 8: Line 155, “22 Å” should be specified as “22 Å in diameter”.

Response: Done.

Comment 9: Line 244, the enzyme that packages the genome in herpesvirus is named “terminase”.

Response: Corrected.

Comment 10: Line 372-381 the method for sample preparation is confusing and seems to be inaccurate, and needs clarification. For example, “repeated freeze-thawing three times” and “the resulting solution was centrifuged through a linear sucrose gradient” suggest that the described protocol was for purification of PRV capsids from pelleted cells. But the authors also mentioned “the culture medium were collected”, and the final product used for cryoEM was described as “the pelleted crude PRV virions”. If culture supernatant was also used to purify intact PRV virions, and combined with PRV capsids from inside the cells, it should be clearly stated and I imagine there should be at least one step of pelleting the virions from the supernatant before it was centrifuged through the density gradient.

Response: We apologize for the confusing description about sample preparation, which was revised and now reads “PK-15 cells were infected with the Bartha strain of PRV at a MOI of 0.01. At 3 days post infection (dpi), culture medium was collected and centrifuged under 1,500 g at 4 °C for 10 min to remove large cell debris. The supernatant was collected and then precipitated by PEG8000 (final concentration of 8% PEG8000). The pellet was resuspended in phosphate buffered saline (10mM PBS, pH 7.4) and further purified by centrifugation through a linear 20% to 50% (w/v in PBS) sucrose gradient at 80,000 g at 4 °C for 4 h. The light-scattering band of viral particles was collected and dialyzed with PBS, then centrifuged through a 20% (w/v in PBS) sucrose cushion at 80,000 g at 4 °C for another 4 h. The pelleted crude PRV viral particles were finally resuspended in PBS.” (Page 20, lines 399-408).

Comment 11: Line 412, “exact” should be “extract”.

Response: Corrected.

Reviewer #2

General opinion and comments:

Reviewer: The manuscript titled "Atomic Structures of Pseudorabies Virus Capsids" by Guosong Wang et al resolved the capsid structures of PRV A and C particles by cryo-EM. By using sub-particle reconstruction strategy, they obtained near atomic structures of both capsids. Then the authors analyzed the structures of MCP and inter/intra interactions between MCPs. The author also further analyzed the structures of triplex, SCP, portal of C-capsid and CATC. Overall, the experimental work is of high quality and it potentially deserves the publication in Nat Commun. However, a number of concerns have to be addressed before the acceptance.

Major points:

Comment 1: The authors claimed that they reported the near-atomic structures of the PRV A-capsid and C-capsid, and illustrate the interaction that occurs between these subunits. While it seems not convincing to define A-capsid or B-capsid by electron microscopy observation. The authors should determine the sedimentation coefficient of these particles by analytical ultracentrifugation to exactly verify A-capsid or B-capsid.

Response: Thanks for pointing this out. As suggested, we performed the analytical ultracentrifugation (AUC) to measure the sedimentation coefficient of viral particles obtained in this study. The result was added in the revised Supplementary Materials (new Fig. S2) which showed that three fractions representing three types of viral capsids with sedimentation coefficient of 750S, 939S and 1,340S, which correspond to PRV A-, B- and C-capsid, respectively, according to the previously studies (Yuan et al., Science 2018;360(6384):eaao7283). The AUC result also revealed that the C-capsid is the dominant component in purified PRV particles, only a small fraction of A-capsids and few B-

capsids were observed. The corresponding description has been added in the revision text and reads: “*The analytical ultracentrifugation was performed to resolve the particle composition of purified PRV particles (Fig S2). Three types of viral capsids with sedimentation coefficient of 750S, 939S and 1,340S, were resolved, which could be ascribed to PRV A-, B- and C-capsid, respectively, according to the previous studies. Of these particles, the C-capsid is the dominant component whereas only a small fraction of A-capsids and few B-capsids were observed (Fig S2).*” (Pages 5-6, lines 93-99).

It is worth mentioning that we purified PRV particles from the supernatant of PRV-infected cell in this study (please see the related materials and methods in Page 20, lines 399-408). Although three light-scattering bands could be seen after linear sucrose density gradient centrifugation, only the lowest band was found to contain a large quantity of viral particles and the upper two bands contain nearly no viral particles. Therefore, we used the viral sample from the lowest band for the AUC analysis and structural study.

Comment 2: Lines 168-175. Compared to HSVs, PRV MCPs exhibit much less inter-subunit interactions between the penton MCP and the adjacent hexon due to the loose and disordered N-lasso and dimerization domains of the penton MCP. Are HSV capsids more stable than PRV capsids? It's necessary to provide functional insights/implications from structural details. Or perhaps correlated with the small-sized genome for PRV and VZV, when compared to HSV and HCMV.

Response: We thank the reviewer for bringing out this interesting point. We found that the overall structure of the PRV capsid resembles that of other α -herpesviruses, especially that of VZV, which similarly harbors looser penton-hexon interactions and triplex-MCP interactions, unlike the numerous disulfide bonds and the strong interactions between penton MCP and the adjacent hexon observed in HSV-1 capsids. Nevertheless, our previous study demonstrated that the capsid stability of VZV is comparable to that of HSV-1, probably due to the smaller genomic DNA (about 125-kb) and therefore lower internal pressure (Wang et al., Nat Microbiol 2020;5(12):1542-1552). However, the genome size of PRV (about 143-kb) was greater than VZV (about 125-kb) but more similar to HSV (about 152-kb). Thus, the stability of PRV capsid should be further investigated as the reviewer's suggestion. So, we performed the thermal stability assay to compare the capsid stability of PRV and HSV-1. The HSV-1 capsid was revealed to withstand high temperature of up to 74 °C, similar to the previous studies (Wang et al., Nat Commun 2018;9(1):3668; Wang et al., Nat Microbiol 2020;5(12):1542-1552), which is significantly higher than that of PRV capsids (~ 53 °C). This result demonstrated that the capsid of PRV may be less stable than that of HSV-1 and VZV due to its looser inter-subunit interactions (compared to HSV-1) and larger genome size (compared to VZV).

We have added the capsid stability result in the new Fig. S10 and the related description added in the Results section with cited literatures now reads “*In order to compare the capsid stabilities of PRV and HSV-1, we performed the thermal stability assay and the result showed that HSV-1 capsid withstood high temperature of up to 74 °C, which is similar to the previous studies, and significantly higher than that of PRV capsids (~ 53 °C) (Fig. S10). This result indicates that the capsid of PRV is less stable than that of HSV-1 possibly due to its weaker inter-subunit interactions*” (Page 9, lines 176-181). We also add the predictive description about the capsid thermal stability in the Discussion section. (Page 16, lines 308-320).

Comment 3: Lines 194-198. Such interactions between the triplex and MCP not only determine the orientation of the triplex, but also render the triplex a hub, facilitating connection of the MCPs and further stabilization of the PRV capsid structure. It seems not right. It's not clear/reasonable that the interactions between the triplex and MCP determine the orientation of the triplex. The triplex lies among three adjacent capsomers, surrounded by at least three buttress domains of three neighbouring MCPs. HOW and WHY such interactions determine the orientation of the triplex??

Response: We agree on that the triplex lies among three adjacent capsomers and is surrounded by at least three buttress domains of three neighboring MCPs and there is no direct evidence for the orientation determination of triplex by the interaction of MCPs. We toned down the related description: “*Such interactions between the triplex and MCP render the triplex a hub, facilitating connection of the MCPs and further stabilizing the PRV capsid structure.*” (Pages 10-11, lines 202-204).

Comment 4: The portal structure in herpesviruses consists of 12 identical pUL6 monomers. But it's very confused why the 12-fold symmetrical portal possesses 5-fold symmetrical tentacle helices in the upper layer? It's weird that the 12 identical pUL6 monomers adopt a 5-fold symmetrical structure, where are the left two? The authors must clarify this point clearly. What evidences were used to support that the 5-fold symmetrical tentacle helices belong to the portal? Why not other protein components?

Response: The 5-fold symmetrical tentacle helices has been previously postulated belonging to the unmodeled residues of the portal clip of pUL6 based on three aspects: 1) the missing residues of the pUL6 clip (residues 308–516) contain predicted long helical stretches interspersed with disordered residues; 2) the dodecameric procapsid portal in P22 phage is known to expose a ‘quasi-fivefold symmetric surface’ at the apex of its clip; 3) the association of the HSV-1 portal with terminase is known to require a leucine zipper in the unmodelled region of the pUL6 clip (Liu et al., Nature 2019;570(7760):257-261). We speculated in this study that the 5-fold symmetrical tentacle helices of PRV may belong to portal based on the above study. We realized that this statement may still controversial even with the current structural information, so we toned down the description: “*The 5-fold symmetrical tentacle helices, which was previously postulated to belong to the unmodeled residues of the portal clip of pUL6, was suggested to bind terminase to mediate the package of viral genome. The portal vertex reconstructed at 4.7 Å resolution under C5 symmetry clearly demonstrated visible Ca bumps and allowed us to build the Ca skeleton model of the tentacle helices...*” (Page 13, lines 250-257).

Comment 5: It has been reported that N-lasso and dimerization domain of EBV and HSV Hexon MCPs have various configurations, however, only one conformation of the Hexon MCPs was found in PRV. Do all these MCPs have the same conformation?

Response: Superimposition of all 15 copies of PRV hexon MCPs from one asymmetric unit (C1-C6, P1-P6 and E1-E3) shows their highly structural similarity, but the P1 and P6 have slightly different conformations in the N-lasso and dimerization domains, which is consistent to the previous observation in HSV-2 and EBV. We therefore introduced a new supplemental figure (new Fig. S8) and related description (Page 8, lines 145-152) to present the conformational variations between hexon MCPs.

Minor points:

Comment 6: Lines 103-104. Many PRV virions as well as naked capsids had been secreted into the culture supernatant. It might not be accuracy. Generally, mature PRV virions with outer membrane are secreted, but naked capsids are not. More probably, the observed naked capsids in supernatant were produced due to the mechanical/physical forces during the purification. Did the authors check the cell supernatant prior to any purification, including concentration?

Response: In this study, we used the cell supernatant of PRV-infected PK-15 cells for virus purification. The cell supernatant was harvested at an extremely late time point (72 hours post PRV infection) and the titer of virus in the supernatant could reach about 5×10^7 pfu/mL. Considering that PK-15 is a susceptible cell line for PRV infection and the late time point for viral harvest, we believed that the naked capsids would exist in the supernatant since a proportion of cell membranes had been ruptured. As pointed out by the reviewer, mature PRV virions are the main

components. However, we can also find the presence of considerable amount of A-capsids. Of course, we agree on that the mechanical/physical forces may also lead to the generation of some naked capsids. To avoid any ambiguity, we now delete the description for the naked capsids which was previously thought to be secreted into the culture supernatant.

Comment 7: Lines 137-139. The models of the asymmetric units of the A- and C-capsids are nearly identical, with a root mean square deviation of 0.298 Å over 26,091 atoms, with the C α positions of the two models aligned (Fig. S6). In general, it should not be like this. Due to the huge pressure produced by genome packaging in C-capsids, at least the portal structures differ radically, including the morphology and location in the capsids (Lokareddy, R.K. et al., Nat Commun 2017; Wang Nan., et.al, Yang Yunxiang.et.al, Protein & Cell 2020).

Response: We acknowledge that the structure of portal vertex is radically different to other five-fold vertexes and we also indeed emphasized this point in our original manuscript (Fig. 6D-E, H-I). But here we just showed the model similarity of A- and C-capsids which were derived from the initial icosahedral reconstructions. By the icosahedral reconstruction, the density of portal vertex (1/12 of all 12 vertexes in one icosahedral capsid) was averaged out and the final density map of C-capsid is highly identical to that of A-capsid with map correlation of 0.965. The resultant model of A-capsid and C-capsid are therefore highly identical except for the absent of CATC on A-capsid. The highly identical icosahedral capsid structures of A- and C-capsids were also revealed in the previously studies (Wang et al., Nat Microbiol 2020;5(12):1542-1552; Sun et al., Nat Commun 2020;11(1):4795). For clarity, we now emphasize in the revised manuscript that the two models are derived from icosahedral reconstruction (Page 7, lines 131-132).

Comment 8: Line 142. 150-kD should be 150-kDa.

Response: Done.

Comment 9: In Fig3g, only two β -sheets can be seen instead of the five β -sheets described in the text.

Response: Sorry for the typo, we have corrected the no. of β -sheets and now reads: “*a stable two-stranded β -sheets and two adjacent loops form interaction network by hydrogen bonding among these segments.*” (Pages 8-9, lines 160-161).

Comment 10: Does buttres of MCP hexon and penton have different conformation?

Response: Yes, the hexon and penton MCPs show some conformational diversity at the buttress domains, we added figures to show this (the new Fig. S8).

Comment 11: In Fig6c, the color bar of ‘genome’ should be changed to green instead of blue.

Response: Corrected.

Comment 12: Does the Fig3e represent the MCP-MCP type II interaction? These are totally inconsistent with the descriptions in line 161-163.

Response: Sorry for the mistake, Lines 161-163 in the original manuscript related to the description of Type II interactions and should cite the Fig. 3f instead of the Fig. 3e. Fig. 3e represents the potential interactions between penton MCP and surrounding hexon MCP, which was cited and described in the next paragraph.

Comment 13: Why the-terminal anchored domain of triplex Tc is disorder?

Response: In the icosahedral reconstructed density maps of both A- and C-capsids, the anchored domains of Tri1 of

Ta, Tb, Td and Te were revealed as short helices except for the Tc, which has a disorder Tril anchored domain. We presume that the Tc anchored domain might interact to the disordered genome and therefore smear upon the icosahedral reconstruction.

Comment 14: The rotation of the triplex Ta adjacent to portal has been reported in HSV-2 and other herpesviruses (Wang Nan et.al Protein & Cell 2020; Hong Zhou. et.al, Cell 2020). It may be universal organization for the assembly of herpesviruses.

Response: We apologize for the negligence and thank the reviewer (reviewer 1 also raised this mistake) to bring this point out. We realized that the triplex Ta surrounding the portal vertex is indeed rotated 120 degree in all available capsid structures of herpesviruses including HSV-1 and HSV-2. This should be a common structural feature of the capsid of herpesvirus. We have revised relative descriptions in Results and Discussion sections, please refer to Page 14, lines 278-281 and Pages 16-17, lines 321-331.

Comment 15: Lines 244-245. The 5-fold symmetrical tentacle-like helix (C5 portal) is supposed to bind terminal enzymes to mediate packaging of the genome. The terminase complex in herpesviruses was reported to form a hexamer (revealed by cryo-EM structure), not a pentamer.

Response: Please see also our response to **Reviewer 2, Comment 4**. The 5-fold symmetrical tentacle helices (postulated as to C5 portal) were speculated to bind terminal enzymes to mediate the package of the genome, which was previously demonstrated in literature (Liu et al., Nature 2019;570(7760):257-261). Here we just described the fact that the tentacle helices contact with terminase complex based on our cryo-EM map, but not to say that the terminase complex is a pentamer. In order to declare our viewpoint more rigorous, we revised these sentences with cited references: “*The 5-fold symmetrical tentacle helices, which was previously postulated to belong to the unmodeled residues of the portal clip of pUL6, was suggested to bind terminase to mediate the package of viral genome. The portal vertex reconstructed at 4.7 Å resolution under C5 symmetry clearly demonstrated visible Ca bumps and allowed us to build the Ca skeleton model of the tentacle helices...*” (Page 13, lines 250-257).

REVIEWERS' COMMENTS

Reviewer #1 (Remarks to the Author):

The authors have thoroughly addressed my concerns, and I retain my enthusiasm in supporting the publication of this paper.

Reviewer #2 (Remarks to the Author):

The authors have answered all my comments and concerns and the manuscript is substantially improved (with errors corrected). I remain extremely enthusiastic about the work and the fact that it should be published.

Response to Editor and Reviewer Comments on the Manuscript NCOMMS-21-28568A

We thank the editor and reviewer for satisfying with majority of changes in our paper. We have addressed the comments and queries of editor fully and revised our paper accordingly. We have edited our paper to comply with the format requirements in the **Nature Communications Author Checklist** and copied the reviewers' comments verbatim in **blue** and typed our responses in **black**.

Reviewer #1 (Remarks to the Author):

Comment: The authors have thoroughly addressed my concerns, and I retain my enthusiasm in supporting the publication of this paper.

Response: Thanks for the reviewer's efforts on this manuscript.

Reviewer #2 (Remarks to the Author):

Comment: The authors have answered all my comments and concerns and the manuscript is substantially improved (with errors corrected). I remain extremely enthusiastic about the work and the fact that it should be published.

Response: Thanks for the reviewer's efforts on this manuscript.